# Study on the Tea Pest Classification Model Using a Convolutional and Embedded Iterative Region of Interest Encoding Transformer

**DOI:** 10.3390/biology12071017

**Published:** 2023-07-17

**Authors:** Baishao Zhan, Ming Li, Wei Luo, Peng Li, Xiaoli Li, Hailiang Zhang

**Affiliations:** 1College of Electrical and Automation Engineering, East China Jiaotong University, Nanchang 330013, China; 3050@ecjtu.edu.cn (B.Z.); 13223076537@163.com (M.L.); 15270030556@163.com (W.L.); lp17344059617@163.com (P.L.); 2College of Biosystems Engineering and Food Science, Zhejiang University, 866 Yuhangtang Road, Hangzhou 310058, China

**Keywords:** tea leaf, image classification, ViT, disease diagnosis

## Abstract

**Simple Summary:**

This paper is mainly based on the tea disease leaves for image classification research, using a combination of convolution, iterative module and transformer in the form of a combination of the traditional convolution for local feature extraction advantage and transformer for global feature extraction potential. The optimal cut size, small sample training ability, anti-interference ability and generalization ability of the model are demonstrated through five sets of experiments respectively. Also at the end of the class activation map visualization was performed to clearly see the model’s classification basis on tea leaves. The results show that the model in this paper is able to accurately capture the location of leaf diseases, which further validates the effectiveness of the model.

**Abstract:**

Tea diseases are one of the main causes of tea yield reduction, and the use of computer vision for classification and diagnosis is an effective means of tea disease management. However, the random location of lesions, high symptom similarity, and complex background make the recognition and classification of tea images difficult. Therefore, this paper proposes a tea disease IterationVIT diagnosis model that integrates a convolution and iterative transformer. The convolution consists of a superimposed bottleneck layer for extracting the local features of tea leaves. The iterative algorithm incorporates the attention mechanism and bilinear interpolation operation to obtain disease location information by continuously updating the region of interest in location information. The transformer module uses a multi-head attention mechanism for global feature extraction. A total of 3544 images of red leaf spot, algal leaf spot, bird’s eye disease, gray wilt, white spot, anthracnose, brown wilt, and healthy tea leaves collected under natural light were used as samples and input into the IterationVIT model for training. The results show that when the patch size is 16, the model performed better with an IterationVIT classification accuracy of 98% and F1 measure of 96.5%, which is superior to mainstream methods such as VIT, Efficient, Shuffle, Mobile, Vgg, etc. In order to verify the robustness of the model, the original images of the test set were blurred, noise- was added and highlighted, and then the images were input into the IterationVIT model. The classification accuracy still reached over 80%. When 60% of the training set was randomly selected, the classification accuracy of the IterationVIT model test set was 8% higher than that of mainstream models, with the ability to analyze fewer samples. Model generalizability was performed using three sets of plant leaf public datasets, and the experimental results were all able to achieve comparable levels of generalizability to the data in this paper. Finally, this paper visualized and interpreted the model using the CAM method to obtain the pixel-level thermal map of tea diseases, and the results show that the established IterationVIT model can accurately capture the location of diseases, which further verifies the effectiveness of the model.

## 1. Introduction

Tea pests are one of the factors affecting the yield and quality of tea. According to incomplete statistics, about a 5% loss in tea production is caused by tea pests every year [1,2]. In actual tea production, tea farmers mostly identify tea pests using their own accumulated planting experience, consulting relevant information on tea diseases, and relying on experienced plant protection experts to conduct on-site inspections [3]. This method has many problems. The first one is the time from when the disease infects tea leaves to when the tea farmers discover the disease and finally take measures to solve the disease. This process often takes a long time, during which the disease will further intensify its spread, and it is often likely that new diseases will appear, leading to missing the optimal treatment period [4]. The second one is the lack of professional knowledge. Tea farmers often misdiagnose diseases based solely on subjective judgment, and it is difficult for plant protection experts to reach remote tea farms on steep mountains [5]. Based on the current problems, manual recognition methods cannot meet the requirements of modern large-scale agricultural planting [6,7].

With the continuous development of image processing and computer vision, traditional human agriculture in China has slowly developed into intelligent agriculture [8] and digital agriculture [9,10] based on the use of computer vision. Machine learning and image processing methods [11,12,13,14] have been widely used in plant disease identification, and the use of computer vision for tea disease identification can not only reduce the human and financial losses caused by manual identification but also improve the quality and yield of tea and increase the economic income of tea farmers. The traditional convolutional space perception range is limited, and the size of the receptive field of operation is fixed. In deep networks, the final perceptual field range is relatively small due to multiple down-sampling operations, which cannot effectively capture global contextual information in images. Self-attention is capable of learning the correlation between features and can interact and integrate features at a global scale. This feature representation capability allows the model to better capture global information and helps to handle long-range dependencies. Therefore, it is of great need to introduce other models beyond convolution to solve this problem. The Transformer [15] is an important innovation in the field of computer vision with excellent global feature extraction capability that can compensate for the deficiency of convolution in feature extraction. Many studies have demonstrated the excellent performance of transformers on vision tasks, such as in image classification [16], target detection [17,18,19], and image segmentation [20,21]. This paper proposes a model that fuses the convolution module, the iterative module, and the transformer module.

The main contributions of this paper are as follows:(1)When testing under complex scenarios, the model remains highly accurate under various complex scenarios.(2)The application of the transformer module to the field of agricultural computers provides more solutions to the problem in the field of agricultural computers.

## 2. Related Work

The development of convolutional neural networks has provided many new solutions for agriculture. Kibriya et al. [22] compared of the effectiveness of tomato leaf pest classification using VGG16 and GoogLeNet models, and the accuracy of VGG16 using the Plant Village dataset was as high as 99%. Atila et al. [23] applied the EfficientNet framework to plant pest leaf classification, and the EfficientNet framework performed best using data from the B5 and B4 models, with accuracy rates of 99.91% and 99.97%, respectively. Liu et al. [24] studied tomato pests and diseases and optimized the feature layer of YOLOv3 with pyramid to achieve multi-scale feature extraction for a faster and more accurate detection process for tomato pests and diseases. Tiwari et al. [25] classified a variety of plant diseases using dense convolution. For plant images with complex backgrounds, the average cross-validation accuracy was 99.58%, the average test accuracy was 99.199%, and the processing time was 0.016 s, indicating that this method not only had high accuracy but also guaranteed the efficiency of processing images. 

The attention mechanism in the Transformer [15] has shown good performance in converting non-structured language into structured language, making the Transformer widely used in the field of natural language. Many researchers have sought to apply the Transformer to the computer field, but due to its unique encoding and decoding system, it is very difficult to apply the Transformer to the computer field. Some scientists associate convolutional neural networks [26] with transformers by expanding the number of channels and space while also attempting to replace traditional convolutional network structures with attention mechanisms. Cordonnier et al. [27] used attention modules to replace traditional convolution, which was far less effective in terms of experimental results than convolutional neural networks. Supported by the continuous efforts of researchers, Dosovitskiy et al. [16] proposed the Vision Transformer (ViT) model in 2020. The process of cutting the input image to a certain size and inputting the position information of the cut image together produces an effect in image classification that is superior to traditional convolutional neural networks. This is a major breakthrough in the field of computer vision.

The traditional convolutional network uses convolutional kernels for feature extraction, which has good performance for local fine-grained feature extraction, but there is still a lot of room for improvement in global coarse-grained feature extraction. Simply increasing the size of convolutional kernels will result in a certain improvement in global feature extraction, but this will lead to excessive parameters and a significant increase in computational complexity, which is not conducive to data training. A transformer compensates for the shortcomings of traditional convolution in global feature extraction. In this paper, convolution and a transformer are combined, and the semantic information and location information in the picture are combined in an iterative way. Using the special coding method in this paper, the original information in the image is retained to the maximum extent for training in the transformer model.

## 3. Image Data

Due to the lack of publicly available authoritative datasets in the field of tea pest control, this paper used manual photography for data collection. Firstly, experts in the field of tea were sought to understand how tea pests are differentiated. Secondly, tea pest-infested leaves were collected from multiple locations from May to June. Finally, the dataset was brought back to the laboratory for manual classification and photography. Tea insect-infested leaves were photographed using a SONY ILME-FX30B camera under natural light, and the following number of samples were collected: 143 images of red leaf spot disease, 113 images of algal leaf spot disease, 100 images of bird’s eye disease, 100 images of gray wilt, 143 images of white spot disease, 100 images of anthracnose disease, 113 images of brown wilt, and 74 images of healthy leaves, totaling 886 images. The pixel resolution of these images was 1024 × 1024, and the pixel size was 200 μm/pixel.

The amount of tea pest data collected in this paper was relatively small, and the leaf positions in the images were random. The characteristics of the seven pests are extremely similar, and there were complex situations such as noise and low brightness in the images. In response to the above situation, this paper performed cropping, filling, and other operations on all images to move the pest infestation part to the center of the image. The data were augmented using flipping, panning, and mirroring. The total number of processed data was 3544, and all the experiments in this paper were pre-trained using the ImageNet dataset.

## 4. Methodology

The IterationVIT model in this paper mainly consists of four modules, as shown in Figure 1. These include (1) the convolution module [28] for local feature extraction, (2) the iteration module, which searches for regions of interest by updating the positions [29], (3) the transformer module, which uses a framework similar to the VIT model, and (4) the classification module.

The purpose of the feature extractor is to extract the feature map of tea leaves. Among them, the convolution module mainly extracts the most original local feature information from the tea leaves, and the iteration module continuously updates the position information using iteration to find the region of interest in the image and ensure the integrity of the semantic information in the image. The transformer module refers to the architecture of the VIT model, and the semantic information obtained with the last iteration is fed back to the convolution feature extraction module, which is passed to the transformer module as the overall input.

### 4.1. Local Convolution Module

The local convolution module mainly uses stacked bottleneck layers with the following basic structure: 1 × 1 convolution kernel (dimensionality reduction), 3 × 3 convolution kernel, and 1 × 1 convolution kernel (dimensionality increase). Among them, the 1 × 1 convolution kernel is used to reduce the number of channels and map the input features to the lower dimensional space, then the 3 × 3 convolution kernel is used to process the features, and finally, the 1 × 1 convolution kernel is used to perform the dimensionality increase operation to restore the number of channels to the original dimension. Stacking bottleneck layers has three major advantages: Firstly, by reducing the number of channels in the middle layer, the number of parameters can be significantly reduced, thus lowering the computational complexity and reducing the risk of overfitting. Secondly, 1 × 1 convolutional kernels can be used to introduce nonlinear transformations and increase the expressiveness of the network. Thirdly, batch normalization and an activation function are added to further improve network performance. As shown in Figure 2, the first and last feature maps are compared using the feature maps extracted at different positions in the bottleneck layer. The local feature extraction of the leaf is very significant: the convolutional neural network first distinguishes the background and the leaf and then carries out leaf feature extraction.

### 4.2. Iterative Region of Interest Encoding Module

An analysis of the VIT model indicates that it categorizes images at a size of 16 × 16 and then maps the cut images and the location information into a set of tensors. However, this approach does not take into account the semantic information in the image, and it also destroys the overall architecture of the image. In order to better preserve the semantic information in the original images, an iterative region of interest encoding method is used in this paper.

The iterative region of interest coding module is an iterative framework. The local feature map obtained with convolution is used as the input of the module, 
$D\in R^{C\times H\times W}$
, where *C*, *H*, and *W* denote the channel dimension, height, and width of the input feature map, respectively. The final output is a set of tensors with location and semantic information, 
$O_{N}\in R^{C\times (n\times n)}$
, where (n 
×
 n) denotes the number of samples into which a picture is segmented and N is the total number of iterations in the module.

As shown in Figure 3, the position information obtained at the completion of each iteration is added to the offset of the previous iteration to update the iteration position. The formula is as follows:
(1)
$$D_{t+1}=D_{t}+M_{t},t\in \{1,\ldots ,N-1\}$$

where 
$D_{t}\in R^{2\times (n\times n)}$
 and 
$M_{t}\in R^{2\times (n\times n)}$
 denote the matrix with position information and the matrix with offset information output, respectively, when iterating t times. For the first iteration, the position information matrix is set with reference to the VIT equidistant spacing to ensure that the iteration covers the whole image. The calculation formula is as follows:
(2)
$$D_{1}^{i}=[\pi_{i}^{y}s_{h}+s_{h}/2,\pi_{i}^{x}s_{w}+s_{w}/2]$$


(3)
$$\pi_{i}^{y}=[i/n]$$


(4)
$$\pi_{i}^{x}=i-\pi_{i}^{y}*n$$


(5)
$$s_{h}=H/N$$


(6)
$$s_{w}=W/n$$

where 
$\pi_{i}^{y}$
 and 
$\pi_{i}^{x}$
 denote the row index and column index of the ith image mapped on the tensor, respectively. 
$s_{h}$
 and 
$s_{w}$
 denote the spacing lengths at the horizontal and vertical coordinates, respectively. The feature map is then input for iterative sampling with the following formula: 
(7)
$$O_{t}=P(F_{t}),t\in \{1,\ldots ,N\}$$

where 
$O_{t}\in R^{C\times (n\times n)}$
 is the initialized sampling information generated with the bilinear interpolation operation at the tth iteration. The initialized sampling information, the sampling information generated with the previous iteration, and the position information in the current sample are input into the encoding layer for encoding, so as to obtain the output information from the current iteration. The formula is as follows:
(8)
$$F_{t}=W_{t}F_{t}$$


(9)
$$X_{t}=O_{t}⊕F_{t}⊕O_{t+1}$$


(10)
$$O_{t}=Transformer(X_{t}),t\in \{1,\ldots ,N\}$$

where 
$W_{t}\in R^{C\times 2}$
 performs a linear transformation operation on 
$F_{t}$
, and 
$W_{t}$
 is shared during the iteration. 
⊕
 denotes a summation operation and is a transformer attention mechanism, which will be explained in detail in the section titled Transformer Encoder Layer.

### 4.3. Transformer Encoder Layer

A transformer is used as the basic module for the iterative region of interest coding module and the Transformer Encoder Layer, which contains the multi-head attention mechanism and the feedforward neural network layer.

(1)Attention mechanism

The attention mechanism consists of two main parts. (1) The transformation layer maps the input sequences 
$X\in R^{n_{x}\times d_{x}}$
 and 
$y\in R^{n_{y}\times d_{y}}$
 to three different sequence vectors using a linear transformation, namely Q (query), K (key), and V (value). n and d are the length and dimensionality of the input sequences, respectively. The calculation formula for each vector generation is as follows:
(11)
$$Q=XW^{Q},K=YW^{K},V=YW^{V}$$

where 
$w^{Q}\in R^{d_{X}\times d^{k}}$
, 
$W^{K}\in R^{d_{Y}\times d^{k}}$
, and 
$W^{V}\in R^{d_{Y}\times d^{k}}$
 are linear matrices, 
$d_{k}$
 is the dimension of query and key, and 
$d_{v}$
 is the dimension of value. The query is projected from X, and the key and values are projected from Y. These two schemes for sequential inputs are called cross-attention mechanisms. Specifically, they can only be referred to as a self-attention mechanism. The self-attention mechanism is widely used in the encoding and decoding modules of the Transformer, and the cross-attention mechanism is only used as a connection point within the decoding module. (2) The attention layer aggregates the query and the corresponding key, and then it aggregates the obtained results with the value again and outputs the update vector. This process is formulated as a unified function:
(12)
$$Attention(Q,K,V)=Softmax(\frac{QK^{T}}{\sqrt{d_{K}}})V$$


The attention weights are generated with a dot product operation between the query and key. In order to maintain gradient stability, a scaling factor 
$\sqrt{d_{K}}$
 is used for scaling, and then the attention weights are normalized using the Softmax function. The normalized weights are assigned to the corresponding elements in value to generate the final output vector.

(2)Multi-headed Attention Mechanism

Due to limited feature subspace, the modeling ability of the single-head attention module is rough. Vaswani et al. [15] proposed a multi-head self-attention mechanism (MHSA), which can be used to improve the performance of the ordinary self-attention layer. The single-headed attention layer limits our capability to concentrate on one or more specific locations without simultaneously affecting attention given to other equally important positions. This is achieved by assigning different representation subspaces to the attention layer. To be more specific, different attention heads use different query, key, and value matrices. Due to random initialization, these matrices can project the trained input vector into different representation subspaces and be processed in parallel with multiple independent attention heads (layers). The resulting vector is aggregated and mapped to the final output. The process of the multi-headed self-attention mechanism can be expressed as:
(13)
$$Q=XW^{Q_{i}},K=YW^{K_{i}},V=YW^{V_{i}}$$


(14)
$$Z_{i}=Attention(Q,K,V),i=1\ldots h$$


(15)
$$MultiHead(Q,K,V)=Concat(Z_{1},Z_{2},\ldots ,Z_{h})W^{0}$$

where i represents the header number, with a number range from 1 to h, 
$W^{0}\in R^{hd_{v}\times d_{model}}$
 represents the output projection matrix, 
$Z_{i}$
 represents the output matrix of each header, and 
$W^{Q_{i}}=R^{d_{k}\times d_{model}}$
, 
$W^{K_{i}}=R^{d_{k}\times d_{model}}$
, and 
$W^{V_{i}}=R^{d_{k}\times d_{model}}$
 are three different linear matrices.

### 4.4. Model Evaluation Metrics

The task in this paper is a classification task, and the evaluation metrics for the classification task are accuracy, recall, precision, F1-measure, AUC, etc.

(1)Accuracy

Accuracy denotes the ratio of the number of correctly classified samples to the total number of samples. The calculation formula is as follows:
(16)
$$Accuracy=\frac{TP+TN}{TP+TN+FN+FP}$$

where TP refers to the number of samples that are actually positive and predicted to be positive; TN refers to the number of samples that are actually negative and predicted to be negative; FN denotes the number of samples that are actually positive and predicted to be negative; and FP denotes the number of samples that are actually negative and predicted to be positive.

(2)Recall

Recall indicates how many positive examples in the sample are predicted correctly. The calculation formula is as follows:
(17)
$$Recall=\frac{TP}{TP+FN}$$

where TP indicates the number of samples that are actually positive and predicted to be positive and FN indicates the number of samples that are actually positive and predicted to be negative.

(3)Precision

Precision indicates how many of the samples predicted to be positive are real positive examples. The calculation formula is as follows:
(18)
$$precision=\frac{TP}{TP+FP}$$

where TP denotes the number of samples that are actually positive and predicted to be positive and FP denotes the number of samples that are actually negative and predicted to be positive.

(4)F1-measure

The F1-measure denotes the summed average of the precision and recall rates. The calculation formula is as follows:
(19)
$$F1=\frac{2\times Precision\times Recall}{Precision+Recall}$$


(5)AUC

AUC is obtained by plotting the average ROC curve under different models and calculating the area under this curve.

## 5. Experiments and Results

This study aims to investigate the performance of the neural network (IterationVIT) after convolution and transformer fusion in the classification performance of tea leaf diseases and to compare it with mainstream convolutional network models (EfficientNet [30], ShuffleNet [31], MobileNets [32], VggNet [33]) and the Transformer model VIT in the field of image classification. All experiments were conducted on a collected tea pest dataset, which contained data on seven tea diseases and one health variable. In order to ensure repeatability of the experiments, a random assignment with a ratio of 8:2 was used, and multiple rounds of testing were conducted. The average of multiple test results was used as the final score to ensure the accuracy and credibility of the tests. Since different initial learning rates and optimizers have a great impact on the training effect of the models, the control variable method was used in this paper. All models were trained with the same SGD optimizer with a learning rate of 0.001 and a batch size of 32, and the training set was trained with a total of 120 epochs using pre-training weight parameters.

### 5.1. Experimental Platform

The operating system for all algorithms in this paper was Ubuntu 20.04, the running environment was based on Python 3.8 under the PyTorch 11.1 framework, and the computer configuration was Intel^®^ PRIME B760M-K D4 with I5 13400, a GPU of NVIDIA Ultra W OC 3060 12 GB, and 32.0 GB of RAM. 

### 5.2. Optimal Size

In order to analyze the optimal patch_size of the transformer module in the IterationVIT model, the tea pest dataset was used as the experimental data, and the patch_size data were set to 8 (patch_8) and 16 (patch_16), respectively. The optimal cutting size of IterationVIT on the tea pest dataset was judged based on different cutting sizes. The obtained results with different parameters are shown in Table 1. The cutting parameters of patch_16 sizes for P, R, F1, and ACC exceeded the cutting parameters of patch_8 size, with F1 being 10.8% higher and ACC exceeding 3.5%. The effects of different cut sizes were visually analyzed using tea pest feature extraction, and the different effects were obtained, as shown in Figure 4. Using a careful observation of the original image, it was found that the pest was located in the lower right corner of the leaf, as indicated using a circular shape. Compared with the six-layer transformer, it was found that in the first layer, patch_8 captured the position of the pest well compared with patch_16, but patch_8 began to extract many positions in the second layer. For example, in the fourth and fifth layers, the head and stem of the leaves were extensively extracted without pests. In the sixth layer, the features of the infested parts were weakened, and the peripheral features of the leaf became the main features for classification. Compared with patch_8, the features of patch_16 in the second layer onwards were constantly approaching the pest features, and finally, the leaf pest position was accurately identified on the sixth layer. In summary, the model had the best classification performance when the input cut size was 16, and the model had better filtering ability for non-disease information on tea leaves and better non-linear selection ability for tea leaf diseases.

### 5.3. Robustness

In actual production, there are many complex scenarios in the detection of tea, such as blurred pictures and strong illumination. This experiment simulated the complex tea detection scenarios, in reality, by processing the test set to varying degrees. This experiment used a comparative approach for the experiment, and three methods were used for processing, namely adding blur, adding noise, and brightness adjustment. The blur processing set two gradients, and the standard deviation of the convolution kernel in the horizontal direction (*X*-axis direction) and vertical direction (*Y*-axis direction) was 5 and 10, respectively. The noise processing set three gradients, and the standard deviation of the Gaussian distribution (sigma) was 0.005, 0.01, and 0.05, respectively, and it was 0.01 and 0.05 for the two gradients for the brightness treatment, with gamma parameters adjusted to 0.5 and 2. Finally, there was a normal control group to analyze the robustness of the model by comparing the experimental results.

The test results for the different processed data are shown in Table 2. It was found that the effect of the normal group was better than that of other groups, where noise had the greatest effect on the model, followed by blur and luminance. In the three gradient experiments on noise, it was found that the accuracy of the model decreased by 10.5% when sigma increased from 0.5% to 1%, and the accuracy of the model decreased by 16.6% when sigma increased from 1% to 5%, which was the largest decrease compared with the other two groups. The lowest AUC value was only 0.5234 when sigma was 5%, indicating that noise had the greatest impact on model recognition. Image blurring had the least effect on the model. When the Gaussian standard deviation was set to 10, the accuracy was 80.1%, higher than 80%, and the AUC was 0.8908. The accuracy of adjusting the brightness of the image to dark and bright was 86.3% and 79.3%, respectively, and the classification accuracy in darker scenes was better than that was brighter scenes.

### 5.4. Small Sample Training

In order to analyze the dependence of the IterationVIT model on the sample size, the tea pest dataset was used as the experimental data, and the number of training samples was constructed at a ratio of 60%, 80%, and 100%. The IterationVIT model and VIT model were trained using the different data samples, and the classification performance of the obtained models was verified using the full test set. The classification accuracy of the obtained IterationVIT and VIT are shown in the red-brown and green bars in Figure 5, respectively. The classification accuracy of the two models increased with an increase in the number of training samples, and the classification accuracy of the IterationVIT model was better than that of the VIT model. When only 60% of the training samples in the training set were used, the classification accuracy of IterationVIT was 80%, which was 8% higher than that of the VIT model. When using the whole training set, the classification accuracy of IterationVIT was 98%, which was 12% higher than that of the VIT model. This result shows that IterationVIT has more obvious advantages than VIT when the sample size is scarce, and IterationVIT has better robustness and is more suitable for small sample classification problems.

Figure 6 shows a further analysis of the IterationVIT model with different training set proportions. The scatterplots illustrate the changes in loss and accuracy in the training and validation process. Panels a and b show the accuracy variation curves for different proportions of training datasets in the training and validation process, where 100% of the training set effects had higher accuracy during the training process than other training processes. Panels c and d show the loss variation curves for the training and validation process of the IterationVIT model, and it can be seen that the loss curves converged quickly and then the process was stable, which reflects the easy training characteristics of IterationVIT. Moreover, as the number of training sets increased at the same time, the training loss changed faster.

### 5.5. Mainstream Model Effect Comparison

In order to verify the classification performance of IterationVIT, this paper used the hardware and software platforms as described above to implement the mainstream network models in the field of image classification for comparison, including EfficientNet [30], ShuffleNet [31], MobileNets [32], and VggNet [33]. All experiments were conducted on a data-enhanced dataset, with a total of eight types of tea leaf data. In order to use hardware devices as much as possible and save training time, we set the batch size to 32. Simulation tests were conducted for each model, and it was found that as the number of iterations increased, the weights in the neural network increased. The curves mostly entered the fitting state gradually at 15 epochs, and at 120 epochs, the curves had completely entered the fitting state. The experiments set the epochs to 20 and saved the optimal model using updating. The eight types of image classification were multi-classification problems, and the experiments used CrossEntropyLoss as the loss function. Based on the advantages of fast training speed, small memory occupation, and adaptability to non-smooth targets, the SGD optimizer was used in the experiment. The specific parameters are shown in Table 3.

Figure 7 shows the accuracy curves for the mainstream model training and validation sets are plotted. The performance capabilities of EfficientNet and ShuffleNet for the tea pest datasets were not significantly different, indicating that adaptive network scaling technology has very good performance on tea pest datasets. The MobileNets network design focuses more on optimizing computational efficiency, with little improvement in accuracy. In addition, the curve for MobileNets indicates that it had significantly lower accuracy than other networks. The accuracy curve for the validation set using the VggNet network was more stable than the other networks, but it performed poorly in terms of accuracy. During the entire training process, the accuracy curves for the training and validation sets of the IterationVIT model were higher than those of other convolutional networks. Compared to the Transformer’s pioneering work in the VIT network, the regions of interest in the IterationVIT model can effectively improve the network’s classification ability.

Figure 8 shows that the recognition ability of each model for different types of images can be observed in the confusion matrix, where 1, 2 … 8 denote the pest types and normal types, respectively. From Figure 8, it can be seen that the IterationVIT network had four pest types with an accuracy of 100%, while the other types of pests had an accuracy of over 92%. For healthy types, there was a slight misclassification rate due to interference from features such as stem and leaf edges in feature extraction, which also occurred in the other models. The recognition rate of the VIT network for several types of pests was mostly between 88% and 100%, and the recognition rates for the first and second types of pests were 74% and 79.1%, respectively. The misrecognition rate for health types was 15%. The EfficientNet network performed best in convolution, with a small difference in the recognition rate for each pest and a false recognition rate of 0 for health types, which were the best-performing among all networks. Compared with the IterationVIT, VIT, and EfficientNet networks, the ShuffleNet, MobileNets, and VggNet models, which identify the types of errors in both pest and health types, appeared disorganized, indicating that these three networks do not have a clear boundary for classification. In general, considering the recognition rate for pest type and the false recognition rate for health type, the IterationVIT network performance was the best.

The objective quantitative metrics obtained for each network are shown in Table 4. Comparing the accuracy of each network, it was found that the accuracy of IterationVIT, EfficientNet, and ShuffleNet was above 90%, with an IterationVIT accuracy of 98%. Comparing the F1-measure for each network, it was found that the IterationVIT and ShuffleNet F1-measure were above 93%, with an IterationVIT value of 96.5%. Comparing the AUC for each network, it was found that all four networks, IterationVIT, VIT, EfficientNet, and ShuffleNet, were above 0.98, with an IterationVIT value of 0.9907. In summary, IterationVIT performed the best.

### 5.6. Model Tests on Public Datasets

Public datasets can verify the generalization ability of the model, and three different public datasets and the dataset used in this paper were compared using the IterationVIT algorithm in order to evaluate the generalization ability of the model. The following table shows the evaluation results for different datasets using the test set after training for 120 epochs.

Table 5 shows the three sets of public plant leaf data and the tea pest leaf data using the IterationVIT model in this paper, the following conclusions can be drawn:

The Swedish Leaf dataset has 15 kinds of plant leaves, and there are 75 pictures for each leaf with a high accuracy and uniform data distribution. The ratio of the training set to the test set was 8:2 for training and testing, and the effect reached 100%. All images were photographed and processed by professionals, completely avoiding the influence of image distribution and image quality on model category classification, resulting in a model classification effect of 100%. The UCI Folio Leaf data, with 32 species of plant leaves, has 20 images for each leaf. The ratio of the training set and test set was 8:2 for training and testing, respectively, and there were more image classification categories compared with the data in this paper. The number of training samples was too small, resulting in a lower model performance than IterationVIT, but the overall effect was also above 90%. The Rice Leaf Diseases dataset contains 3 types of rice pest images, with 40 images per image. The difficulty in classification was comparable to that when using data in this paper. The ratio of the training set to the test set was 8:2 for training and testing, respectively, and the final result was 97.3%, which is different from the results in this paper by 1.5%. In summary, the IterationVIT model has good generalization ability.

### 5.7. Visual Interpretation of the Model

In order to confirm the validity of the IterationVIT model, the class activation maps (CAM) for the prediction set on the tea dataset were obtained. CAM is an important tool to observe and understand the feature maps, which can show the importance of different pixel positions on the model output results with different colors. If the feature map obtained from the last convolutional layer in the IterationVIT model is 
K
, which has 
$n_{ch}$
 channels, and the weight between the channels 
$K_{i}$
 and the predicted values 
$w_{i}$
 in the last fully connected layer in the model is, then the calculation process of CAM can be expressed as:
(20)
$$CAM(K)=\sum_{i=1}^{n_{ch}}w_{i}K_{i}$$


The obtained class activation diagram was scaled to the size of the original diagram and superimposed on the original diagram as a heat map, and the obtained results are shown in Figure 9. The parts of the image that are close to red in color are important criteria for the model to judge the image category, the yellow-green area indicates a certain influence on the classification result, and the blue-purple area has no obvious effect on the classification result.

As shown in Figure 9, the insect infestation areas in (a), (b), and (c) of the image were concentrated around the tea leaves. The insect infestation areas in the tea leaves had obvious places of concern, and there were also small areas of concern in the background around the insect infestation location. This is because the model is more sensitive to the area around the pest in the process of feature extraction, but the small area does not have an impact on the final classification results of the model. The pests in images (d) and (e) were concentrated in the interior of tea leaves, and the model accurately captured the location of the pest. In large areas of insect infestation, the model accurately extracted the most critical information, which improved the model’s accuracy. The pests in images (f) and (g) were scattered in various parts of tea leaves, and the area of concern in the model occupies most of the tea leaf area, among which there were also some obvious areas of concern, which is an important basis for the final judgment of the pest class. Compared with the previous situations, large-scale pests pay more attention to cluttered information such as background, which has a certain impact on the accuracy of the model. There is still room for further optimization in this regard.

## 6. Conclusions

In this paper, an IterationVIT model based on convolution, an iterative model, and transformer fusion is proposed for the classification of tea pest images. The proposed IterationVIT model takes advantage of convolution for local feature extraction and then obtains the semantic information matrix of the images using iteration. The matrix with semantic information is input into the transformer together with the image information and finally classified using the classification layer. After small sample testing, it was found that compared to VIT, IterationVIT is more suitable for small sample training. When only 60% of the training samples were taken out, the classification accuracy was 80%, which was 8% higher than for VIT. After robustness testing, it was found that the model testing effect reached over 80% in cases of slight noise addition, blur addition, and highlight addition. Compared with the mainstream models, it was found that IterationVIT outperforms EfficientNet (accuracy: 0.906), ShuffleNet (accuracy: 0.916), MobileNets (accuracy: 0.761), VggNet (accuracy: 0.804), and VIT (accuracy: 0.860) models in terms of F1-measure and accuracy, with an F1-measure of 0.965 and accuracy of 0.98. Using four sets of experiments, it was well confirmed that the IterationVIT model can accurately identify tea disease categories. Finally, the generalization validation was conducted on three sets of plant leaf public datasets, and the experiments indicated that the IterationVIT model has good generalization performance. There are three more priorities for future work. Firstly, amplify pest types. Secondly, the trained model should be deployed on professional agricultural equipment. Thirdly, agricultural equipment should be put into use in real life.

## Figures and Tables

**Figure 1 biology-12-01017-f001:**
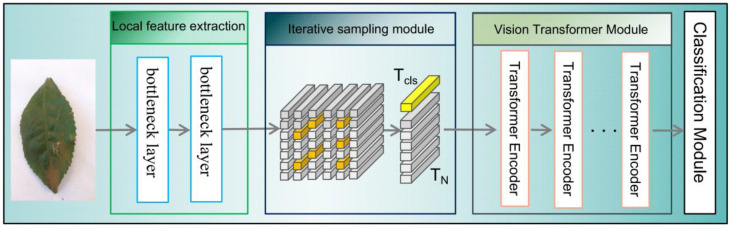
Flowchart showing the overall architecture of the convolution module, iteration module, transformer module, and classification module.

**Figure 2 biology-12-01017-f002:**
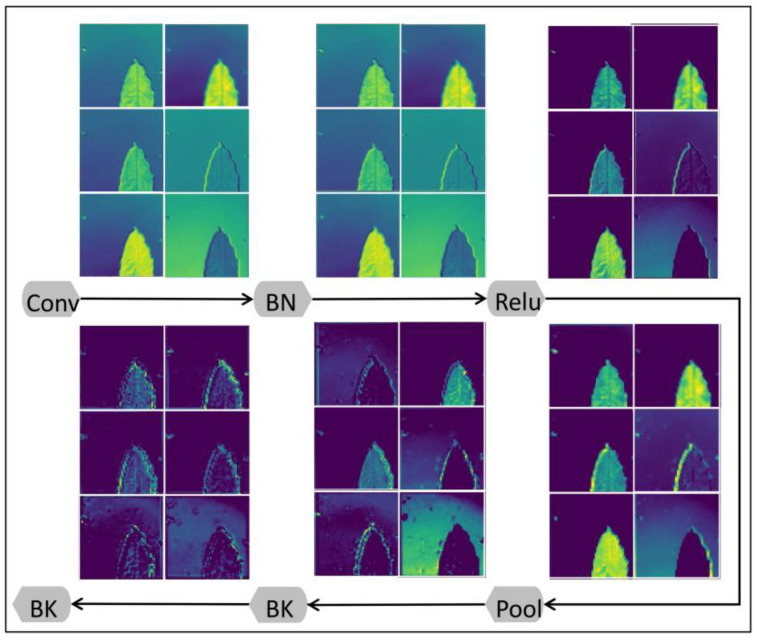
Flowchart showing the process of extracting tea leaf feature maps using the local convolution module: Bk: bottleneck layer, Conv: convolution, BN: batch normalization, Relu: activation function, Pool: pooling.

**Figure 3 biology-12-01017-f003:**
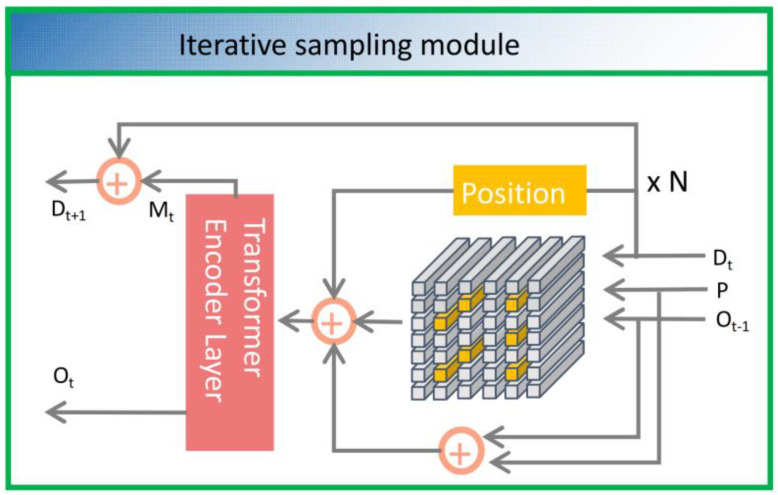
An internal architecture diagram showing the iterative region of interest encoding module. Among them, D_t_ represents the input of the image, P represents the initialized sampling position, O_t_ represents the current layer offset information, O_t−1_ represents the previous layer offset information, and Transformer Encoder Layer represents the module with the attention mechanism. By manipulating the position information of the input D_t_, P, and O_t_, the current region of interest can be obtained. Then, the Transformer Encoder Layer module is used to encode the current region of interest to obtain information for the next iteration. After repeating this process N times, the region of interest in the image can be obtained.

**Figure 4 biology-12-01017-f004:**
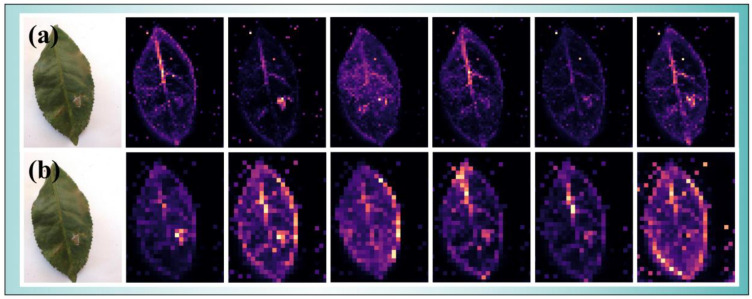
Images showing the visualization results from tea pest feature extraction: (**a**) for the case with a patch_size of 16 and (**b**) for the case with a patch_size of 8.

**Figure 5 biology-12-01017-f005:**
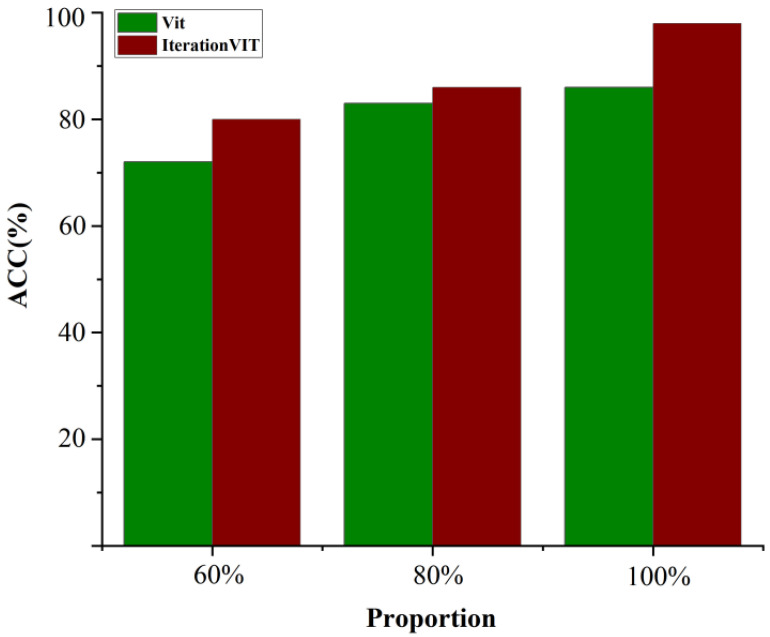
Bar graph showing the effect of different training sample sizes on the classification performance of VIT (green) and IterationVIT (red-brown).

**Figure 6 biology-12-01017-f006:**
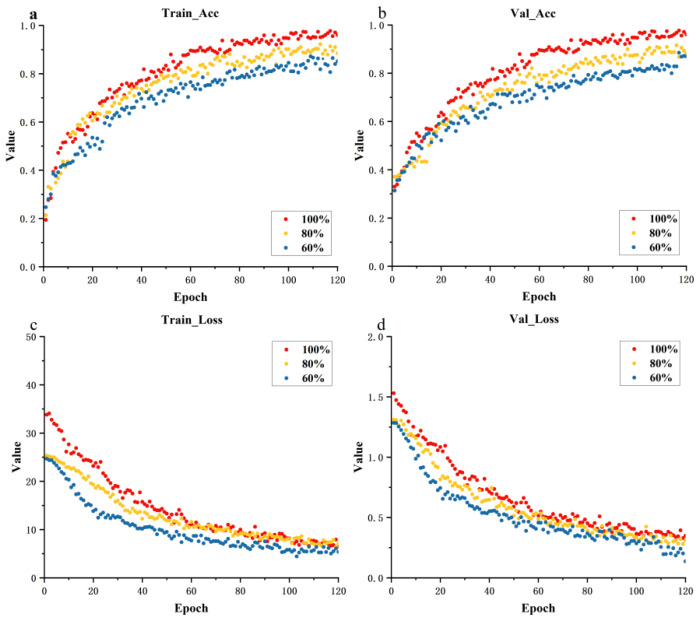
Scatterplots showing the training curves using different ratios of the training dataset and validation dataset, where 100%, 80%, 60% refer to the ratios of the training set and validation set. (**a**): training set accuracy change curve, (**b**): validation set accuracy change curve, (**c**): training set loss change curve, (**d**): validation set loss change curve.

**Figure 7 biology-12-01017-f007:**
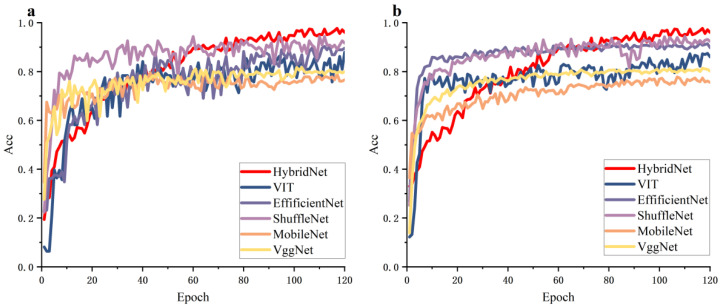
Mainstream model training and validation set accuracy curves. (**a**) training set accuracy curve, (**b**) validation set accuracy curve.

**Figure 8 biology-12-01017-f008:**
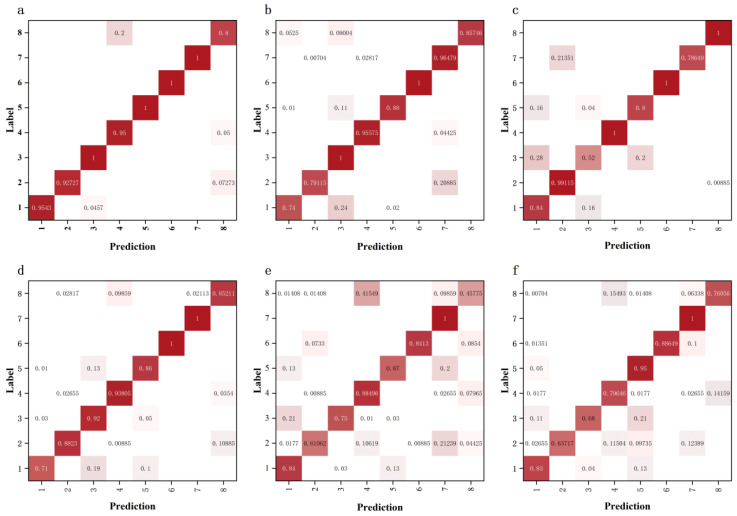
Confusion matrices for different networks using the tea pest image dataset. (**a**) IterationVIT, (**b**) VIT, (**c**) EfficientNet, (**d**) ShuffleNet, (**e**) MobileNets, and (**f**) VggNet.

**Figure 9 biology-12-01017-f009:**
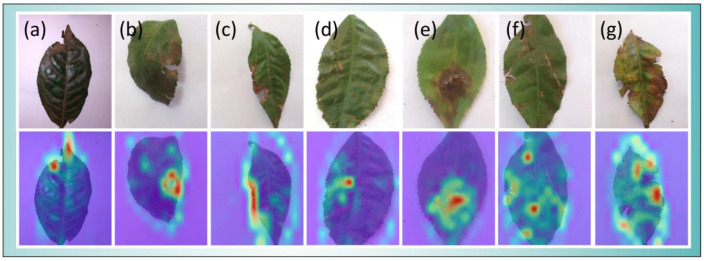
Images showing the IterationVIT category activation plot obtained using the tea data validation set. (**a**–**c**): location of the infestation is at the edge of the leaf, (**d**,**e**): the location of the infestation is inside the leaves, (**f**,**g**): the infestation spreads throughout the leaves.

**Table 1 biology-12-01017-t001:** Classification results under different image cutting sizes.

Patch Size	Precision (%)	Recall (%)	F1-Measure (%)	Accuracy (%)	AUC
8 × 8	95.6	77.7	85.7	94.5	0.9868
16 × 16	98.8	94.4	96.5	98	0.9907

**Table 2 biology-12-01017-t002:** Accuracy of the test set under different treatments.

Model	Precision (%)	Recall (%)	F1-Measure (%)	Accuracy (%)	AUC
IterationVIT	0.988	0.944	0.965	0.980	0.9907
Blur—5	0.860	0.820	0.840	0.850	0.9546
Blur—10	0.810	0.790	0.800	0.801	0.8908
Noise—0.005	0.830	0.780	0.804	0.810	0.8638
Noise—0.01	0.690	0.720	0.705	0.705	0.8223
Noise—0.05	0.560	0.450	0.499	0.539	0.5234
Brightening—0.5	0.870	0.810	0.839	0.863	0.9377
Brightening—2	0.780	0.830	0.804	0.793	0.9281

IterationVIT: No processing, Blur—5, Blur—10: blur processing, the standard deviation of the convolution kernel was 5 and 10, respectively. Noise—0.005, Noise—0.01, Noise—0.05: noise processing, the standard deviation of the Gaussian distribution was 0.005, 0.01, and 0.05, respectively, Brightening—0.5, Brightening—2: gamma: plus brightness processing, the parameters were adjusted to 0.5 and 2, respectively.

**Table 3 biology-12-01017-t003:** Training parameters.

Parameter	Value
Optimizer	SGD
Loss function	CrossEntropyLoss
Batch size	32
Epochs	120
Learning rate	0.001

**Table 4 biology-12-01017-t004:** Comparison of the evaluation metrics for each network using the tea pest image dataset.

Model	Precision (%)	Recall (%)	F1-Measure (%)	Accuracy (%)	AUC
IterationVIT	0.988	0.944	0.965	0.980	0.9907
VIT	0.823	0.838	0.830	0.860	0.9868
EfficientNet	0.846	0.951	0.895	0.906	0.9807
ShuffleNet	0.946	0.924	0.935	0.916	0.9826
MobileNets	0.678	0.715	0.696	0.761	0.7925
VggNet	0.715	0.795	0.753	0.804	0.8649

**Table 5 biology-12-01017-t005:** Classification results in four public datasets.

Datasets	Precision (%)	Recall (%)	F1-Measure (%)	Accuracy (%)
Swedish Leaf	1	1	1	1
UCI Folio Leaf	0.900	0.925	0.908	0.925
Rice Leaf Diseases	0.970	0.943	0.952	0.973
Our Data	0.980	0.944	0.965	0.988

## Data Availability

The data presented in this study are available on request from the corresponding author. The data are not publicly available due to subsequent studies by other authors need to be further researched.

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
