# Peer review of "Study on the Tea Pest Classification Model Using a Convolutional and Embedded Iterative Region of Interest Encoding Transformer"

_biology, 2023, doi:10.3390/biology12071017_

Round 1
Reviewer 1 Report
Thank you for the interesting paper
using k-folds is recommended to consolidate the results
The paper’s application topic is interesting (detecting regions of interest and classifying the tea leaves’ infection type), so probably this is an important application in the biology realm. From the computer vision and ML perspective, the main contribution is what the authors presented as the iterative region of interest encoding, and the idea is quite interesting and innovative. However, although they authors exerted good effort to explain the " iterative region of interest encoding", the figures and the wording can be enhanced to make it more comprehensive. For example, the sampling of the positions in the feature map extracted from the inputs through the first convolution model and feeding them to the transformer is not that clear. Also Fig3 looks great but neither the graph, nor the caption give the complete picture of how it works.Also, the contribution can be much better if the authors can publish their source code. Even better, if possible, publishing the data would also be a big contribution. Offering the source code and the data would allow researchers to regenerate the results and build over it to advance the with idea.
Author Response
您好,感谢您的宝贵意见,这是我的回复。

Reviewer 2 Report
Suggestions and Issues Identified:
Lines 56-58:
The sentence "Traditional convolution performs well in local feature extraction, but there are still shortcomings in global feature extraction" appears incomplete. There needs to be a more comprehensive explanation on what specific issues arise in global feature extraction. The structure of the Iteration ViT mentioned later acts as a bottleneck for local feature extraction, but what advantages does it offer from a global feature extraction perspective?
Lines 99-109:
Although there is an explanation given in this section, it would be beneficial to provide more detailed insights into the advantages of global feature extraction (computational costs, speed, performance, etc.). A comparative figure outlining the structure of CNN and Iteration ViT would also be helpful.
Line 110:
It would be beneficial to include sample images of various diseases from the IMAGE DATA for a more complete understanding.
Lines 313-314, 331:
The text states that "The cutting parameters of patch _ 16 sizes in P, R, F1, and ACC exceeded the cutting parameters of patch _ 8 size, with F1 being 10.8% higher and ACC exceeding 3.5%". However, in Table 1, the performance of the patch size 16*16 seems to be lower than 8*8. This contradiction needs clarification.
Line 394:
The reliability of the results presented in Figure 5 is questionable. Is the same outcome achieved upon repeated trials? More information on this would be helpful.
Line 468:
There appears to be a typo; "EffificientNet" should be "EfficientNet". It would also be beneficial to include comparison analyses with other models utilizing ViT (e.g., ViT-e, ViT-G).
None
Author Response
您好,感谢您的宝贵意见,这是我的回复。

Reviewer 3 Report
Overall, the study proposes a tea disease diagnosis model called Iteration VIT that combines convolution and iterative transformer to address the challenges of recognizing and classifying tea images due to random lesion location, high symptom similarity, and complex background. The model is trained and evaluated using a dataset of 3,544 tea leaf images, achieving high classification accuracy and outperforming mainstream methods such as VIT, Efficient, Shuffle, Mobile, Vgg, etc. The study also tests the robustness of the model by introducing blur, noise, and highlighting effects to the test images and still achieving satisfactory classification accuracy. Additionally, the model is compared to mainstream models using a reduced training set and shows improved performance with the ability to analyze fewer samples. Finally, the study visualizes and interprets the model using the CAM method, generating pixel-level thermal maps of tea diseases to validate the accuracy of disease localization. While the study presents promising results and contributes to the field of tea disease management, there are some major concerns to consider:
1. The study mentions using a dataset of 3,544 tea leaf images for training and evaluation. However, it is important to assess the representativeness of this dataset and ensure that it captures a wide range of tea diseases and variations in leaf characteristics. If the dataset is biased or lacks diversity, it may impact the generalizability of the proposed model to real-world scenarios.
2. The study reports high accuracy and F1 measure for the Iteration VIT model. However, it would be beneficial to have a more comprehensive evaluation, including other performance metrics such as precision, recall, and possibly a confusion matrix. These additional metrics provide a more nuanced understanding of the model's performance and its ability to correctly classify different tea diseases.
3. The study claims that the Iteration VIT model outperforms mainstream methods, but it would be valuable to compare the proposed model with more recent state-of-the-art models in the field of computer vision. Providing a comparison with the most up-to-date approaches would help to assess the competitiveness and advancement of the proposed model.
4. To further establish the effectiveness and generalizability of the Iteration VIT model, it is important to validate it using external datasets that were not used during training or mentioned in the study. This validation would provide insights into how well the model performs on unseen data and whether it can be reliably applied in real-world tea disease management scenarios.
5. The study mentions visualizing and interpreting the model using the CAM method, generating pixel-level thermal maps of tea diseases. While this is a valuable step in understanding the model's decision-making process, it would be beneficial to provide more detailed explanations and analysis of these thermal maps. How accurately do the thermal maps represent the actual disease locations? Are there any limitations or potential sources of error in the visualization process?
6. Evaluation metrics (i.e., accuracy, recall, etc.) are well-known and have been used in previous studies such as PMID: 35767281, PMID: 28285094. Therefore, the authors are suggested to refer to more works in this description to attract a broader readership.
7. The conclusion mentions that the proposed IterationVIT model outperforms mainstream models in terms of F1-measure and accuracy, but it does not provide specific values for these metrics. Without precise numerical values, it becomes difficult to evaluate the extent of improvement and compare the performance of the IterationVIT model with other models in a meaningful way.
8. The study mentions small sample testing and robustness testing, but it does not elaborate on the evaluation methodology or provide details on the size of the test sets used. Additionally, it states that the model's effect reaches over 80% in cases of slight noise addition, blur addition, and highlight addition, but it does not specify the baseline accuracy without these effects. Without a clear baseline or comparison, it is challenging to assess the significance of the reported results.
9. The study suggests that the trained model can be deployed on professional agricultural equipment and used in real-life scenarios. However, it does not provide any evidence or results to support this claim. Real-world deployment and validation are crucial to ensure the practical applicability and effectiveness of the proposed model, but this aspect is not addressed in the study.
10. The study briefly mentions that the IterationVIT model can accurately identify tea disease categories, but it does not discuss the model's generalization ability or its performance on unseen data. It is important to assess how well the model performs on new or unseen tea pest images that were not included in the training or evaluation sets. Without such analysis, it is uncertain whether the model's performance will remain robust and accurate in real-world situations.
11. The study mentions viewing the confusion matrix and finding that only one class of pest identification has an accuracy rate below 90%. However, it does not provide any insights or explanations regarding the reasons behind misclassifications or the implications of these results. A more detailed discussion on the model's limitations and potential sources of error would enhance the conclusion and provide a more comprehensive understanding of the model's performance.
12. Uncertainties of models should be reported.
13. Quality of figures should be improved.
Overall, English writing and presentation style should be improved.
Author Response
您好,感谢您的宝贵意见,这是我的回复。

Round 2
Reviewer 3 Report
My previous comments have been addressed.